# Holistic Photoprotection, Broad Spectrum (UVA-UVB), and Biological Effective Protection Factors (BEPFs) from *Baccharis antioquensis* Hydrolysates Polyphenols

**DOI:** 10.3390/plants12050979

**Published:** 2023-02-21

**Authors:** Yéssica A. Monsalve-Bustamante, Félix López Figueroa, Julia Vega, Bruna Rodrigues Moreira, Miguel Puertas-Mejía, Juan C. Mejía-Giraldo

**Affiliations:** 1Grupo de Investigación en Compuestos Funcionales, Facultad de Ciencias Exactas y Naturales, Universidad de Antioquia UdeA, Calle 70 No. 52-21, Medellín 050010, Colombia; 2Departamento de Ecología y Geología, Instituto Universitario de Biotecnología y Desarrollo Azul (IBYDA), Universidad de Málaga, Campus Universitario de Teatinos s/n, 29071 Málaga, Spain; 3Phycology Laboratory, Botany Department, Federal University of Santa Catarina, Florianopolis 88049-900, SC, Brazil; 4Grupo de Estabilidad de Medicamentos, Cosméticos y Alimentos, Facultad de Ciencias Farmacéuticas y Alimentarias, Universidad de Antioquia UdeA, Carrera 50A No 63-85, Medellín 050010, Colombia

**Keywords:** *Baccharis antioquensis*, dermocosmetic, natural sunscreen, photoprotection, UVA–UVB

## Abstract

Overexposure to solar radiation has become an increasingly worrying problem due to the damage to the skin caused by ultraviolet radiation (UVR). In previous studies, the potential of an extract enriched with glycosylated flavonoids from the endemic Colombian high-mountain plant *Baccharis antioquensis* as a photoprotector and antioxidant was demonstrated. Therefore, in this work we sought to develop a dermocosmetic formulation with broad-spectrum photoprotection from the hydrolysates and purified polyphenols obtained from this species. Hence, the extraction of its polyphenols with different solvents was evaluated, followed by hydrolysis and purification, in addition to the characterization of its main compounds by HPLC–DAD and HPLC–MS, and evaluation of its photoprotective capacity through the measurement of the Sun Protection Factor (SPF), UVA Protection Factor (UVAPF), other Biological Effective Protection Factors (BEPFs), and its safety through the cytotoxicity. In the dry methanolic extract (DME) and purified methanolic extract (PME), flavonoids such as quercetin and kaempferol were found, which demonstrated antiradical capacity, as well as UVA–UVB photoprotection and prevention of harmful biological effects, such as elastosis, photoaging, immunosuppression, DNA damage, among others, which demonstrates the potential of the ingredients in this type of extract to be applied in photoprotection dermocosmetics.

## 1. Introduction

In recent years, people have a greater awareness of the harmful effects of ultra-violet radiation (UVR) on the skin, caused by damage to DNA, proteins, enzymes, and various cellular structures, which leads to the development of cancer, premature skin-aging (wrinkles, dryness, dilation of blood vessels, and loss of collagen and elastin), or alterations in the immune system. As a consequence of the direct damage of UVR on these biomolecules, type I and type II photosensitized reactions or oxidative stress resulting from the accumulation of ROS (reactive oxygen species) occur [1,2,3,4,5]. Due to the above, the growing use of sunscreens in the general population has been noted, but concerns about safety and the impact on health and the environment have also increased due to the constant use of photoprotectors [6,7,8].

There are different action spectra related to the effects of UVR on the health. Action spectra is defined as a measure of the effectiveness of the radiation in carrying out a photobiological process. This is expressed as a plot of the reciprocal of the radiant exposure required to produce a given effect at each wavelength, which describes the relative effectiveness of UVR in producing a biological response for a given wavelength, being specific for a certain effect. Therefore, the action spectra most used for the determination of photoprotection factors are erythema and persistent pigment darkening (PPD) for the determination of sun protection factor (SPF) and UVA protection factor (UVAPF), respectively. Other important action spectra can be elastosis, photoaging, immunosuppression, or DNA damage, among others (Figure 1). In this sense, de la Coba et al., 2019 suggested the use of different action spectra to calculate the Biological Effective Protection Factors (BEPFs); these are analogues to the SPF or UVAPF, but with other action spectra. A mathematical forecast of the potential protective impact of a cosmetic ingredient against the harmful effects of UV radiation is made using the photoprotection factors, which are defined as the amount of time a person can be exposed to the sun before experiencing the relevant biological injury (UVR) [9,10,11,12,13].

These UV biological effects can be counteracted with the continuous use of topical sunscreens made from inorganic UV filters such as TiO_2_ and ZnO and/or organic UV filter sunscreens. However, these ingredients have presented several problems, as is the case with inorganic filters in which the interaction with visible light is highly depends on the particles sizes, as particle sizes around 10 µm result in white pigmentation, which is not sensorially accepted in the cosmetic industry. Nevertheless, to avoid this effect, when reducing the particle size to nanometers, they become practically translucent. However, the main problem with the use of these inorganic nanoparticles is their photocatalytic activity, since ZnO-nano and TiO_2_-nano are semiconductors and after UVR absorption, some valence band electrons are promoted to the conduction band, generating a hole-electron pair. This leads to the formation of ROS (O_2_^−^, OH•), which behave as second messengers or oxidizing agents of several biomolecules, such as proteins, polyunsaturated fatty acids, and DNA, generating cytotoxicity and genotoxicity [14,15,16]. On the other hand, some organic filters have presented dermatological problems, such as dermatitis, allergies, photo-contact allergies, or behave as endocrine disruptors, in addition to presenting local or systemic toxic effects. As a consequence of their degradation by photolysis, systemic absorption or their degradation products occurs [1,5,8,17,18].

Hence, there is a need to develop sunscreens through the evaluation of new sources of sunscreens from natural sources, which help in the design of improved commercial formulations of sunscreens, with broad-spectrum absorption capabilities, that are effective, safe, and offer a good sensory experience. In this sense, the methanolic extract of *Baccharis antioquensis,* a plant species endemic to Colombia that grows in high mountain ecosystems in tropical areas, showed excellent antioxidant and broad-spectrum photoprotective properties associated with the presence of chlorogenic acid and glycosides of quercetin and kaemferol, which gives it great potential to be used as a natural source in cosmetic products; however, the extract proved to be photounstable [19,20,21,22]. Therefore, in this study, we evaluated the extraction of polyphenols from *B. antioquensis* using different solvents, followed by their hydrolysis and purification. In addition, chemical characterization was performed by HPLC–DAD and HPLC–MS, and the photoprotective capacity was evaluated through the measurement of the SPF, UVAPF, and other BEPFs, and their safety through the cytotoxicity.

## 2. Results

### 2.1. Extraction of Metabolites from Dried Plant Material of Baccharis antioquensis

In the first assay, sequence extractions were carried out using solvents of different polarity in increasing order. Significant changes in the color of the extracts were shown. It was observed that the greatest quantity of pigments, such as chlorophylls and carotenoids, were extracted with solvents of lesser polarity such as hexane, dichloromethane, and ethyl acetate (Appendix A). In the UV–Vis absorption spectra (Figure 2), it was observed that the polarity of the solvents directly affected the extracted metabolites. The non-polar solvents (hexane and dichloromethane) mainly extracted metabolites that absorb in the visible region, as can be seen in the color of the extracts (Appendix A), with minimal absorption in the UVA–UVB region. In the spectrum of the extract with ethyl acetate, a solvent of medium polarity, co-extraction with little selectivity was observed. Apolar pigments and polar metabolites of the polyphenol type, which are out of our interest, were obtained from this extraction and discarded. On the other hand, the absorption spectra of the extracts in ethanol and methanol (Figure 2) presented absorption bands throughout the UVB range and much of the UVA range, and no absorption in the visible region of the spectrum, without presenting qualitative differences between them. In this sense, the relative absorption coefficients (RAC) values (Figure 3) for the extracts in methanol and ethanol presented the best absorption characteristics at 290, 310, 340, and 380 nm. However, the methanol extract presented the best RAC and yield values. The wavelengths 290, 310, 340, and 380 nm were selected for the following reasons: (i) the four wavelengths cover the UVA and UVB ranges; (ii) 290 nm is the lower limit of the UVB range; (iii) 310 nm is the maximum peak of the effective erythematous spectrum (the λ where the greatest erythema occurs is between 306–310 nm); (iv) 340 nm is considered the maximum wavelength where the plateau of the solar spectrum is reached in the UVA range, which is the inflection point of the PPD action spectrum and it is close to the maximum peak of the effective PPD spectrum (350 nm); and (v) 380 nm is 10 nm above the critical wavelength (λc: 370 nm), which is the essential value to declare broad spectrum protection.

In assay 2, after the removal of lipids and pigments with hexane, no significant differences were observed in the UV–Vis spectra on the extraction with different concentrations of ethanol in the hydroalcoholic mixtures. On the other hand, for the alkaline extract, a perceptible change in the absorption spectrum was observed, with respect to the hydroalcoholic extracts (Figure 4). 

Regarding the extraction yields (Figure 5A), lower yields were observed for the alkaline extraction, while the 50% hydroalcoholic mixture presented a significantly higher yield compared to the other treatments. Additionally, the RAC of the alkaline extractions were the lowest with a statistically significant difference with respect to the hydroalcoholic extracts that were higher and did not present a difference between them (Figure 5B). The best TPC and DPPH (EC50) values were obtained when methanol was used in assay 1 and ethanol (50%) in assay 2 as the extraction solvent (Table 1). Comparing the obtained data, was observed that the most promising treatment corresponded to the degreasing process with dichloromethane and subsequent extraction with methanol by maceration, which was implemented to obtain a dry methanolic extract (DME) in assay 3 (Appendix A). The DME was subsequently hydrolysate and purified (PME). The absorption spectrum for DME showed greater absorbance in the UVB region and in part of the UVA region, while the absorption spectra for PME showed absorbance in both regions (UVB–UVA) (Figure 6), which gives it an increase in photoprotection in the UVA region and, therefore, improves broad-spectrum protection.

### 2.2. Chemical Characterization

#### 2.2.1. HPLC–DAD Analysis

In the chromatographic profile of DME and the spectra of the main peaks (Appendix A), it was shown that these compounds are responsible for the absorption in the entire UVB region and part of the UVA region, with maximum absorption between 325 and 353 nm. In addition, it was qualitatively observed how the sum of these spectra constitutes an additive effect, which gives the DME the absorption capacity in the UVB region and much of the UVA (Figure 6, green line). In Appendix A, the chromatographic profile of PME and the UV spectra of the main peaks are shown, where it was found that the compounds present in the purified extract have a higher absorption in the UVA region. 

Additionally, in the chromatogram of the flavonoid standards, rutin presented a retention time (RT) of 2.180 min and a UV spectrum with a maximum at 353 nm, and quercetin presented an RT of 11.045 and an absorption maximum at 370 nm (Appendix A). In this sense, in the DME chromatographic profile, rutin can be presumptively identified with a RT of 2.197 and quercetin with a RT of 11.352, both with spectral characteristics similar to the standard, which is corroborated by the UPLC–ESI–MS/MS analysis below.

#### 2.2.2. UPLC–ESI–MS/MS Analysis

The main components of the DME were identified: peak 2 with a RT of 5.4 corresponds to 3-O-caffeoylquinic acid (chlorogenic acid), with a parent ion at 355.1033 m/z and its fraction 163.0397 *m*/*z* that corresponds to a rearrangement after the loss of glucose. Peaks 5 and 7 (RT: 6.6 and 7.2, respectively) correspond to caffeoylquinic acid hexose, with its parent ion at 517.1351 m/z and its fraction at 163.0397 *m*/*z*, the same as in chlorogenic acid. Peak 8 with a RT of 7.3 belongs to rutin (quercetin 3-O-rhamnopyranosyl-(1→6)-glucopyranose), where its fractions of 465.1033 and 303.0508 m/z were evidenced by the loss of pyranose and glucose, respectively. Peak 10 (RT: 8.0) was identified as kaempferol 3-O-rhamnopyranosyl-(1→6)-glucopyranose with various fragmentations as the ion 163.0393 m/z, which corresponds to the caffeoyl fragment after loss of quinic acid. Peak 11 (RT: 8.0) belongs to quercetin-3-O-(4‴-O-caffeoyl)-rhamnopyranosyl-(1→6)-galactopyranose (Appendix A).

Likewise, the flavonoid aglycones quercetin and kaempferol were identified in the PME, according to their chromatographic and spectral characteristics. Thus, in peak 1 with an RT of 2.8 and a molecular ion of 291.2666 *m*/*z*, it was identified as epicatechin/catechin. In addition, a molecular ion *m*/*z* 465.0821 corresponding to Quercetin-3-glucoside resulted from the partial hydrolysis of rutin with xylose loss. In turn, the ion *m*/*z* 465.0821 has a product ion at *m*/*z* 303.0507 *m*/*z* that corresponds to the aglycone quercetin due to the loss of glucose as a neutral fragment. Additionally, several quercetin derivatives were observed, since several peaks in PME showed a fragment of 303.05 *m*/*z*. On the other hand, Peak 9 (RT: 9.1) was identified as monoglycosylated kaempferol with a parent ion of 481.1135 *m*/*z*, peak 10 and 11 with RT of 9.3 for both, belonging to the quercetin aglycone, and peak 12 (TR: 9.7) to the kaempferol aglycone (Appendix A). 

The superimposition of the chromatograms (DME and PME) showed that the hydrolysis was effective and mainly associated with the absence of most of the peaks in the chromatogram of the hydrolyzed extract (Figure 7). In addition, it was clearly observed that in the DME, polar compounds predominate with an RT below 8 min and clear behavior of polar glycosylated compounds (red chromatogram), and in the PME chromatogram, more apolar compounds (aglycones) predominate, which are more retained by the stationary phase of the column, predominantly with an RT above 8 min (blue chromatogram). Figure 8 shows the different structures of the main compounds identified in the UPLC–ESI–MS/MS analysis (QTOF positive mode) for DME and PME.

### 2.3. Cellular Cytotoxicity of PME

To check the preliminary safety of PME, its cytotoxic activity was measured in vitro in a keratinocyte cell line (HaCaT cells). Thus, the proliferation of these cells was treated with different concentrations (3.9–60 µg/mL) of PME for 72 h and their cytotoxicity was subsequently measured (Appendix A). As a result, a reduction in the viability of HaCaT cells was found in a concentration-dependent manner after 72 h of treatment with an IC_50_ of 44.2 µg/mL. 

### 2.4. Evaluation of Photoprotection and Photostability

For the evaluation of photoprotection and photostability, cosmetic emulsions containing 10% (*w*/*w*) of the extract or a mixture of extracts were prepared (Appendix A). For this, the extracts were solubilized in propylene glycol and incorporated into the formulation in a 10% (*w*/*w*) ratio. For the cosmetic formulation, it was checked that each component was of natural origin with a specific and additive function in skin care. Additionally, it was taken into account that the incorporation of the different extracts that had different physicochemical characteristics did not affect the stability of the emulsion.

The extracts DME and PME and mixtures thereof were evaluated (Table 2). It was found that the highest SPF*_in vitro_* value corresponded to the formulations containing DME and the best result for UVAPF*_in vitro_* was shown by PME. In this sense, it is important to emphasize that a good sunscreen must have a broad spectrum of UVA–UVB protection. For this to be met, the UVAPF factor must be at least one third of the SPF. Thus, it was found that the formulations containing PME and DME:PME (70:30) achieved this parameter; however, this ratio for the DME and DME:PME mix (80:20) did not meet the parameter due to the large difference in absorption in the UVB region with respect to a lower absorption in the UVA region (Figure 6). Likewise, the best UVA/UVB ratio was for the PME with a value of 0.95, which would give it five stars in the Boots Stars^®^ convention, due to similar absorption values in the two UVR regions. For the critical wavelength in all formulations, the FDA and ISO recommendations are met, which must be equal to or greater than 370 nm for a photoprotection product to be considered broad spectrum. Considering that the best value was for PME at 388.8 ± 0.4 nm and the lowest value was for DME at 374.4 ± 0.6 nm, all formulations can be considered broad spectrum according to FDA parameters. 

Photostability was obtained after exposing the formulations to a radiation dose of 650 W/m^2^ for two hours. As a result, the formulation that presented the greatest loss of efficacy after irradiation was DME. On the other hand, although the formulation with PME presented low values of SPF*_in vitro_* compared to DME, the photostability of the formulation with PME proved to be quite good, with %SPFeffective 97.9% and %UVAPFeffective 100%, since a formulation is considered stable when it retains 80% of the efficacy after the irradiation time. Other photoprotection parameters did not show significant changes with prolonged irradiation. Furthermore, it was noted that the greatest loss of efficacy of the formulations generally occurred in the first 30 min.

On the other hand, the potential efficacy of DME and PME was evaluated to prevent elastosis, photoaging, immunosuppression, lipid peroxidation, DNA damage, photocarcinogenesis, and singlet oxygen production, through BEPFs (Table 3). We found that DME mainly prevents photoaging with a factor of 8.5 ± 1.1, immunosuppression by 10.5 ± 1.4, DNA damage by 10.5 ± 1.3, carcinogenesis by 9.5 ± 1.1, and singlet oxygen production by 9.6 ± 1.3; for this last parameter, DME presents the same capacity as PME with a value of 7.9 ± 0.8 (*p* < 0.05). However, PME exhibits better protection against photoaging (11.3 ± 1.4), lipid peroxidation (7.6 ± 0.8), and elastosis of 10.9 ± 1.5 with no significant difference compared to the protection shown by the positive control. Additionally, PME prevents, to a lesser extent, DNA damage and photocarcinogenesis.

The positive control presented high BEPFs, highlighting the protection against photoaging, singlet oxygen production, and immunosuppression, while the base cream showed no protection through BEPF parameters. Therefore, it was observed that the extracts of natural origin, that are the object of this work, demonstrated a high potential to prevent the biological effects of UVR studied thanks to their polyphenolic content, which allows them to play different roles in photoprotection.

## 3. Discussion

Regarding the yield of the extraction, in Figure 3A, the percentages for each solvent were shown; low yields are observed for the extracts of the apolar solvents (hexane and dichloromethane), which were mainly extracted metabolites that absorb in the visible region, as can be seen in the color of the extracts (Appendix A), with minimal absorption in the UVA–UVB region; these are not of interest to our research. Additionally, the maximum absorbance of these two extracts were presented in characteristic spectral regions (640–700 nm) for compounds of the type of chlorophylls a and b and carotenoids [23]. Therefore, the use of these solvents facilitated the removal of interfering compounds such as lipids and pigments. In assay 2, in the alkaline extract, a perceptible change in the absorption spectrum was observed with respect to the hydroalcoholic extracts (Figure 4), which is because the phenols in solution are usually influenced by pH due to the formation of resonance in the benzene rings, producing altered conjugation compared to the original compound, which depends largely on the structure of the phenol [24]. Additionally, it was noted that the best extraction yields were obtained with polar solvents and their mixtures with water (ethanol and methanol), in accordance with the widely reported chemical nature of polyphenols [25,26]. When assessing the relationship between the extraction yield and the RAC, it was concluded that methanol presented the best extraction yields as well as the RAC values (Figure 3). Hydrolysis and purification of DME (PME) presented changes in the absorption spectrum with a shift towards the UVA region, and showed a higher polyphenol content and a DPPH radical scavenging capacity that is comparable with commercial antioxidants. 

The EC_50_ DPPH tests indicated the antiradical capacity of extracts from *B. antioquensis* to donate hydrogens or electrons and neutralize free radicals compared to ascorbic acid control. This showed that the antiradical capacity was increasing with the enrichment of polyphenols from DME, compared to PME, with the latter presenting values equal to ascorbic acid (*p* < 0.05). With the previous tests and the chemical characterization (Section 2.2), it was shown that the hydrolysis and purification managed to isolate the largest amount of flavonoid aglycones in each step of the treatment, which correlates with the good antiradical capacity. According to the analyses carried out by HPLC–DAD and UPLC–ESI–MS/MS (QTOF positive mode), the corresponding protonated molecular ion [M + H]^+^ and its most important fragments were determined for the majority compounds; these results are consistent with those reported by Mejía-Giraldo et al., 2016 [22], suggesting the presence of flavonoids, confirmed by the mass spectrometry results.

However, the hydrolysis process could be carried out fractionally and consecutively, using a shorter reaction time in each fractionation and extracting the hydrolyzed extract in each portion of time and continuing with the hydrolysis process of the same sample. This procedure would help ensure that the reaction medium does not affect the aglycones produced by hydrolysis, preventing them from being degraded by remaining for a long time under the reaction conditions, and consequently, increasing the hydrolysis yield.

Regarding the cell toxicity, the immortalized human keratinocyte cell line HaCaT has been extensively employed in pharmacological and mechanistic research of prospective skin medicines, including polyphenols [27]. Likewise, it was compared with the IC_50_ of 50.9 ± 3.3 μg/mL for DME, determined by Mejía-Giraldo et al., 2016 [22], for which values close to each other were found for both extracts in terms of the minimum amount required to inhibit 50 % of cell viability. Therefore, it could be concluded that the hydrolysis of DME did not significantly increase its cytotoxicity. On the other hand, Rossi et al., 2020 [27] found an IC_50_ of around 500 μg/mL for both a polyphenolic extract and quercetin, while Mendoza-Meza & España-Puccini, 2016 [28], found an IC_50_ of 91.85±12.05 mg/mL in HACAT cells for their phenolic extract. Likewise, Milanezi et al., 2019 [29] did not report any significant change between the control and quercetin, up to a concentration of 12.5 μg/mL, on L929 fibroblasts cells, which is an insufficient concentration to notice a change, as found by other authors. When comparing the previous results with those obtained for the DME and PME extracts, these were considered weakly cytotoxic. 

The SPF is based on the effectiveness of a sunscreen to prevent erythema, which is mainly related to UVB radiation, while the UVAPF is the index that allows protection from UVA and is related to the prevention of the various biological effects caused by this radiation, including PPD [30,31]. However, there are other effects of UVR on the skin, such as elastosis, photoaging, immunosuppression, photocarcinogenesis, oxidative stress, and DNA damage, among others. De la Coba et al., 2019 [13] suggested effective biological protection factors (BEPF’s), which can show the potential protective effect of an extract or a substance against the harmful biological effects mentioned above [32]. 

The values of the different BEPFs (Table 3) must be interpreted as SPF. As we explained above, BEPFs are an indicator of photoprotection from different biological effects. Thus, the value of 13.9 in the positive control is a protection factor higher than that from the extracts, 10.5 and 6.1, as SPF 50 means higher protection against erythema than SPF values of 15 or 30. All values of BEPFs for the positive control (sunscreen SPF 16) are higher than the protection of the extracts DME or PME, except in the case of elastosis, which is the same in the case of PME extracts and the positive control. The DME extract presented higher photoprotection than the PME extract against biological effects mediated by UVB radiation (immunosuppression, DNA damage, and photocarcinogenesis (NMSC)), due to DME extracts presenting a maximal absorption in the UVB region of the spectra. Whereas PME extracts presented higher photoprotection than DME extracts against biological effects mediated by UVA radiation, such as elastosis, photoaging, and lipid peroxidation, due to the PME extract presenting with maximal absorption in the UVA region spectra. Thus, a cream containing both DME and PME could be considered a broad-band UV screening product since it can protect against biological effects mediated by both UVB and UVA [13,33].

In this sense, promising photoprotection parameters were obtained for most of the formulations. The different photoprotection factors found in the extracts evaluated demonstrated that compounds of natural origin have varied activities, with multiple applications. In addition, the great potential of PME can be appreciated, since this extract has a low solubility in water, which allows us to infer that it will not be easily lost by perspiration or immersion in water, giving it a potential to be used in water or sweat resistant formulations with very good substantivity characteristics. 

The photostability capacity evidenced for the formulation with PME may be a product of the concentration of the resonant cores of the polyphenols, as a consequence of the hydrolysis and purification treatment, due to the fact that in this procedure, the sugars were eliminated, to which the photolytic instability of flavonoid glycosides can be attributed, which was presented in the formulation containing DME. It should be noted that for the formulations with mixtures containing DME and PME, the increase in the proportion of PME gave the formulation greater photostability and increased UVAPF. 

Thus, the best performance in terms of photoprotection could be attributed to DME, but the highest photostability was presented by the PME formulation, which also showed very good photoprotection parameters. When comparing the formulation containing 10% natural extract with commercial photoprotectors containing an approximately 15–30% mixture of chemical and physical filters to achieve high levels of photoprotection with SPF between 25–50, remarkable results were obtained.

## 4. Materials and Methods

### 4.1. Reagents and Plant Material

Dichloromethane, methanol, ethanol, hexane, ethyl acetate, hydrochloric acid, anhydrous sodium sulfate, sodium carbonate, Folin-Ciocalteu reagent, acetonitrile, formic acid, and ascorbic acid were obtained from Merck Chemical Supplies (Damstadt, Germany). Glyceryl monostearate, propylene glycol, medium chain triglycerides, soy lecithin, jojoba oil, lanolin, ceteareth 12, mannitol, polysorbate 80, cethiol, and glycerin were obtained from Cosmética Nataural Casera (Málaga, España). Gallic acid, rutin, and quercetin standards, 3-(4,5-dimethylthiazol-2-yl)-2,5-diphenyltetrazolium bromide, were purchased from Sigma Chemical Co. (St. Louis, MO, USA).

The plant material was collected on 26 April 2019 in Yarumal (Llanos de Cuivá), Antioquia-Colombia, at 2730 m above sea level (geographical coordinates: 6°49050.6″ N; 75°29029.0 W) under the access contract to genetic resources and their derived products No. 252, Resolution 0399. The biological material was washed with plenty of water and dried at room temperature for 14 days, protected from light, and ground to a uniform powder in an electric mill (Hamilton Beach, 80350R) until dry plant material (DPM) was obtained, which was stored in the dark.

### 4.2. Establishment of the Conditions for the Extraction of Metabolites from Dried Plant Material of Baccharis antioquensis

Assay 1. Sequence extractions of DPM was carried out using solvents of different polarities, as follows: hexane, dichloromethane, ethyl acetate, ethanol, and methanol. The DPM was exposed to the solvent by maceration for 24 h, this procedure was carried out three times. Then, the DPM obtained was dried at room temperature and continued with the following solvent in order of polarity, taking a 1:20 (p/v) DPM-solvent ratio for each extraction. All extractions were performed in triplicate (Appendix A).Assay 2. The DPM was treated with hexane 1:20 (*w/v*) for 2 h with ultrasound exposure (Branson, 2510E-MT), to remove lipids compounds. Then, the solvent was removed by centrifugation, decantation, and drying at room temperature for 24 h. Subsequently, extractions of DPM were made with three different solvents; this procedure was carried out twice with a DPM–solvent ratio of 1:20 (p/v) using a different composition of solvents, as follows: ethanol, ethanol:water (50:50), and solution alkaline 3.0 M. The DPM extraction technique was carried out first by mechanical disruption and subsequent exposure to ultrasound for 2 h at a controlled temperature of 25 °C with the corresponding solvent. Then, it was centrifuged, decanted, and the solvent was removed by rotary evaporation at 35 °C and/or subsequent lyophilization depending on solvent type. All extractions were performed in triplicate.Assay 3. The DPM was degreased with dichloromethane in a 1:10 (*w/v*) ratio for 24 h. This treatment was carried out three times, and the DPM resulting was dried at 35 °C in an oven (Memmert, UF55) for 24 h. Subsequently, the extraction by maceration with methanol was realized in a 1:10 *w/v* ratio for 24 h; this procedure was repeated five times and the methanolic extract obtained was dried in a rotary evaporator (IKA, RV10 basic) at 40 °C. The dry methanolic extract (DME) was stored at room temperature and protected from light. All extractions were performed in triplicate.

In each assay carried out, its performance, absorption spectra, and relative absorption coefficients (RAC) at 290, 310, 340, and 380 nm were determined.

### 4.3. Hydrolysis of Methanolic Extract of Baccharis antioquensis

DME was dissolved in methanol (50% *v*/*v*) in a 1:40 (*w*/*v*) ratio. Subsequently, it was adjusted with hydrochloric acid to the required concentration and placed in the oven at the temperature and time defined for each treatment. Hydrolysis was stopped by cooling the mixture with ice for 5 min. Next, it was centrifuged at 3500 rpm for 30 min, the supernatant was decanted, and the precipitate obtained was redissolved in distilled water, followed by liquid–liquid extraction with ethyl acetate. After liquid–liquid extraction, the organic phase was dried with anhydrous Na_2_SO_4_, filtered through a 0.45 μm membrane, and dried in the oven at 35 °C. The hydrolyzed methanolic extract (HME) obtained was weighed and subsequently redissolved in methanol for subsequent tests. All experiments were performed in triplicate.

### 4.4. Purification of the Hydrolyzed Extract of Baccharis antioquensis

HME purification was carried out using preparative reverse phase chromatography using C18 silica gel as the stationary phase and methanol as the mobile phase. The fraction with the highest RAC and yield was collected, thus obtaining the hydrolyzed and purified methanolic extract (PME). Subsequently, it was dried in the oven at 35 ºC for storage, chemical characterization, photoprotection, and photostability tests.

### 4.5. Chemical Characterization

#### 4.5.1. UV–Vis Absorption Spectra and Relative Absorption Coefficients (RAC) in the UVA–UVB Range

The absorption spectrum was obtained in the UV–visible range (200–700 nm). Relative absorption coefficients (RAC) were determined at 290, 310, 340, and 380 nm, expressed as absorbance (A)/mg of dry extract/mL [34]. A spectro-photometer Evolution 60S was used for all spectrophotometric measurements (Thermo Fisher Scientific, Inc., Shanghai, China).

#### 4.5.2. Total Phenol Content

Total phenols content (TPC) in all the extracts was determined according to the modified Folin–Ciocalteu (FC) colorimetric method [34,35]. TPC was calculated from calibration curves for gallic acid ($y=115.42x+0.0093,R^{2}=0.9997)$ and expressed as milligrams of gallic acid equivalents (GAE). All samples were prepared and analyzed in triplicate.

#### 4.5.3. Evaluation of the Inhibition Capacity of DPPH

The antiradical capacity of all extracts was determined using the DPPH• stable radical assay, as free radicals [36]. The decrease in absorbance was determined at 515 nm. The initial concentration of DPPH• ([] DPPH) in the reacting medium was estimated from the calibration curve $y=0.0099x-0.014,\;R^{2}=0.9998$. The remaining percentage of DPPH• was determined based on the concentration of dry extract (mg/mL). Antiradical activity was reported as the effective concentration (EC50 = steady state test concentration/DPPH t = 0 concentration). The outcomes were contrasted with the antiradical efficacy of the reference standards BHT and ascorbic acid.

#### 4.5.4. High Performance Liquid Chromatography HPLC–DAD

Analysis of DME and PME components (0.1 mg/mL in methanol) was performed using the Agilent Technologies 1260 high performance liquid chromatography (HPLC) instrument (Agilent, Palo Alto, CA, USA), equipped with a variable wavelength UV detector (VWD-G7114A) and quaternary pump (G7111A). The separation was performed by injecting 20 μL of sample into a Waters^TM^ Symmetry C18 column (5 µm, 4.6 × 150 mm) using an elution gradient with a mobile phase composed of eluent A (acetic acid in acetonitrile 0.1% (*v*/*v*)) and eluant B (acetic acid in water 0.1% (*v*/*v*)). The gradient program started with 20% A (2 min), 20–10% A (4 min), 10% A (4 min), 10–20% A (4 min), and 20% A (2 min), with a flow rate of 1.0 mL/min, followed by washing and reconditioning of the column. The parameters of retention time and UV–Vis spectra were compared with rutin and quercetin standards.

#### 4.5.5. UPLC–ESI–MS/MS Analysis

The identification of the DME and PME components (0.1 mg/mL in methanol) was carried out using an UPLC coupled to a Bruker Impact II QTOF mass spectrometer (Bruker, Billerica, MA, USA). The separation was performed in reverse phase on a Symmetry^®^ column (4.6 mm × 75 mm, particle size 3.5 μm; Waters, Ireland) at constant temperature (30 °C). The injection volume was 10 μL. The mobile phase was composed of eluent A (formic acid in water 0.1% (*v*/*v*)) and eluent B (acetic acid in acetonitrile 0.1% (*v*/*v*)). The gradient program was as follows: 50% B (0–0.5 min), 50–95% B (0.5–7 min), 95% B (1 min), 95–50% B (8–9 min), and 50% B (9–11 min), with a flow rate of 0.4 mL/min. The temperature of the samples was 15 °C. Ionization was performed in positive ion ESI–MS/MS mode, in a mass/charge ratio range between 50–2000 amu.

### 4.6. Cytotoxicity Test

For the cell viability assay, keratinocyte cells (HaCaT) were incubated (6000 cells/well) with extract concentrations of 3.9 to 60 µg/mL PME. The experiment was performed in a 96-well microplate for 72 h (37 °C, 5% CO_2_ in a humid atmosphere). The proliferation of this cell line was estimated using the MTT assay (3-(4,5-dimethylthiazol-2-yl)-2,5-diphenyltetrazolium bromide), using RPMI as culture medium [37]. Briefly, a volume of 10 μL of the MTT solution (5 mg/mL in phosphate buffered saline) was added to each well and the plates incubated at 37 °C for 4 h. The yellow tetrazolium salt of MTT is reduced by metabolically active viable cell mitochondrial dehydrogenases to form insoluble purple formazan crystals. The formazan was dissolved by adding HCl-isopropanol (150 μL of 0.04 N HCl-2-propanol) and measured spectrophotometrically at 550 nm (Micro Plate Reader 2001, Whittaker Bioproducts, Los Angeles, CA, USA). Relative cell viability was expressed as IC_50_, which corresponds to 50% viable cells compared to untreated cells [37]. Four samples were included in each experiment for each concentration tested. Measurements were carried out in triplicate.

### 4.7. Development of the Emulsion-Type Cosmetic Formulation

The developed formulation was prepared as an oil-in-water (O/W) emulsion, where the oily phase made up 25% of the emulsion, using raw materials reported as being of natural origin in the entire formulation (Appendix A). The aqueous and oil phases were prepared independently and heated to 70 °C. The oily phase was then incorporated into the aqueous phase and homogenized at 10,000 rpm. For this preparation, the percentage, function, and characteristics of each ingredient were considered.

### 4.8. In Vitro Evaluation of Photoprotective Efficacy

The method described by ISO 24443:2012 [30] was followed, with some modifications. The photoprotective capacity was evaluated in vitro by diffuse reflectance spectroscopy with an integrated sphere. The formulations were spread on polymethyl methacrylate (PMMA) plates applying 1.3 mg/cm^2^ evenly and distributed uniformly over the whole surface using a finger. Plates were left to stand at 25 ± 2 °C for 30 min in the dark and subsequently UV transmission (290–400 nm) was measured. In vitro photoprotection efficacy was calculated according to the following parameters: UVB efficacy by estimating sun protection factor (SPF), and UVA efficacy by the UVAPF, UVA/UVB ratio, and critical wavelength (λ_c_). Four plates were prepared by formulation and nine different points per plate were measured for each sample.

### 4.9. Photostability of Sunscreen Formulations

The photostability study was performed according to the method used by Couteau et al., 2021 [38,39,40,41,42]. Plates were prepared following the same steps as described in the above section, were exposed for 2 h to irradiation (measurements were taken every 30 min) in simulated solar conditions at 650 W/m^2^ (UVB: 2 W/m^2^, UVA: 48 W/m^2^, Vis: 550 W/m^2^, IR: 50 W/m^2^) in accordance with the global solar spectral irradiance. All in vitro parameters characteristic of the photoprotection of the formulations (SPF, UVAPF, UVA/UVB ratio, and critical wavelength (λ_c_)) were measured before and after irradiation. The photostability was expressed as the percentage of effectiveness of SPF*_in vitro_* (% SPF_eff_) and UVAPF (%UVAPF_eff_), which were estimated based on Equations (1) and (2), respectively. Four plates were prepared by formulation and nine different points per plate were measured for each sample.
(1)$$\%SPF_{eff}=\frac{SPF_{in\;vitro\;}after\;irradiation}{SPF_{in\;vitro}\;before\;irradiation}\;\times \;100$$
(2)$$\%UVAPF_{eff}=\frac{UVAPF_{in\;vitro\;}after\;irradiation}{UVAPF_{in\;vitro}\;before\;irradiation}\;\times \;100$$

### 4.10. In Vitro Evaluation of BEPF’s

The BEPFs were calculated for the spectra of elastosis, photoaging, immunosuppression, lipid peroxidation, DNA damage, photocarcinogenesis, and singlet oxygen production, according to Equation (3):(3)$$\%\;Protection_{BEPF}=\frac{\int_{290}^{400}BAS\left(\lambda \right)I\left(\lambda \right)d\lambda }{\int_{290}^{400}BAS\left(\lambda \right)I\left(\lambda \right)10^{-A\left(\lambda \right)}d\lambda }$$
where BAS(λ) is the relative efficacy of UVR to produce each biological action at wavelength λ (see Figure 1); I(λ) is the spectral irradiance received from the UVR source at wavelength λ; A(λ) is the absorbance at wavelength λ and d(λ) wavelength step (1 nm) [13].

The transmittance of all plates were determined in a UV–vis 2600 spectrophotometer (Shimadzu, Tokyo) coupled to an integrating sphere.

### 4.11. Statistical Analysis

All experiments were performed at least three times and the results were expressed as the mean ± SD. All data were evaluated by single factor and multifactor analysis of variance (ANOVA) followed by Tukey’s tests, when applicable, to determine the difference between means using Statgraphics Centurion 18 (Statgraphics Technologies Inc., 2018) and Microsoft Excel. *p* values less than 0.05 (*p* < 0.05) were considered significant.

## 5. Conclusions

A cosmetic formulation with broad-spectrum photoprotective properties was developed from hydrolyzed and purified polyphenols obtained from the leaves of the *B. antioquensis* plant, for which different flavonoid aglycones corresponding to the nuclei of quercetin, kaempferol, and chlorogenic acid were characterized; these are responsible for the antioxidant and photoprotective capacity that PME presents. In addition, it was demonstrated, by calculating the BEPFs, that the formulations containing DME and PME effectively absorb UVR, generating potential protection against the harmful biological effects associated with it when converting these extracts into useful bioingredients in dermocosmetic applications.

## Figures and Tables

**Figure 1 plants-12-00979-f001:**
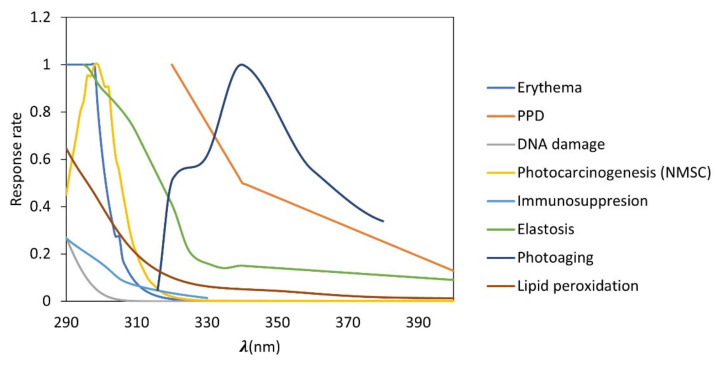
Biological action spectra related to different effects of ultraviolet (UV) radiation. UVB: erythema, DNA damage, photocarcinogenesis (NMSC; non-melanoma skin cancer), immunosuppression, UVA: persistent pigment darkening (PPD), elastosis, photoaging, and lipid peroxidation.

**Figure 2 plants-12-00979-f002:**
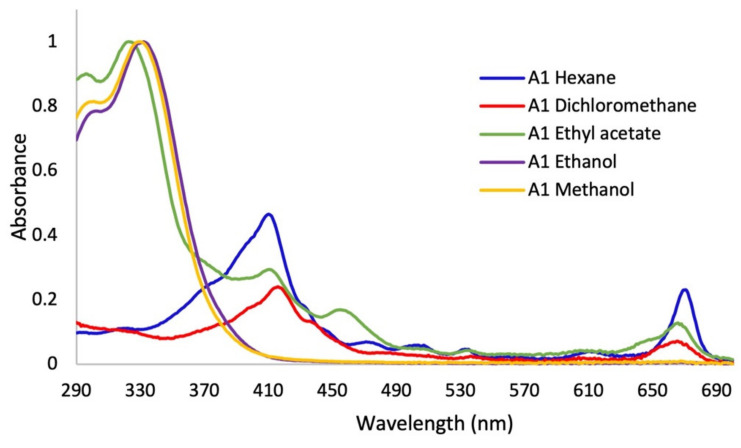
UV–Vis absorption spectra of *Baccharis antioquensis* sequential extractions in different solvents (Assay 1).

**Figure 3 plants-12-00979-f003:**
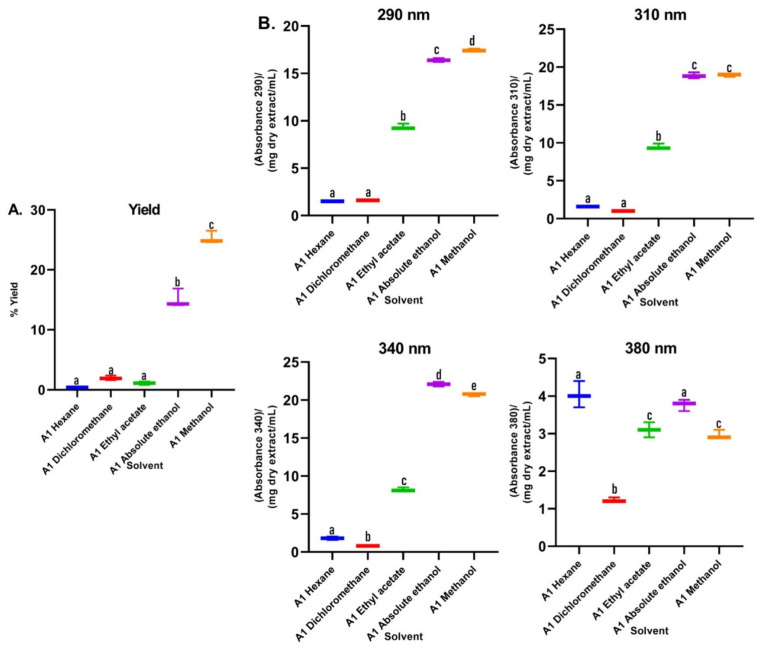
(**A**) Yields of *B. antioquensis* sequential extractions in different solvents. (**B**) Relative absorption coefficients (RAC) of extracts of *B. antioquensis* (Assay 1). Values with different letters are significantly different (*p* < 0.05).

**Figure 4 plants-12-00979-f004:**
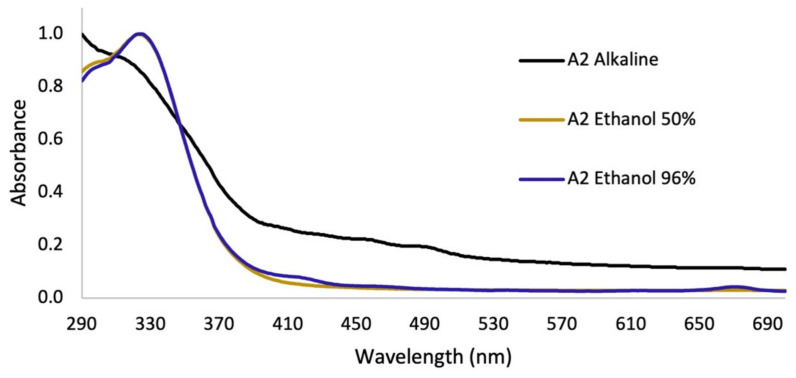
UV–Vis absorption spectra for extractions of *B. antioquensis* Assay 2.

**Figure 5 plants-12-00979-f005:**
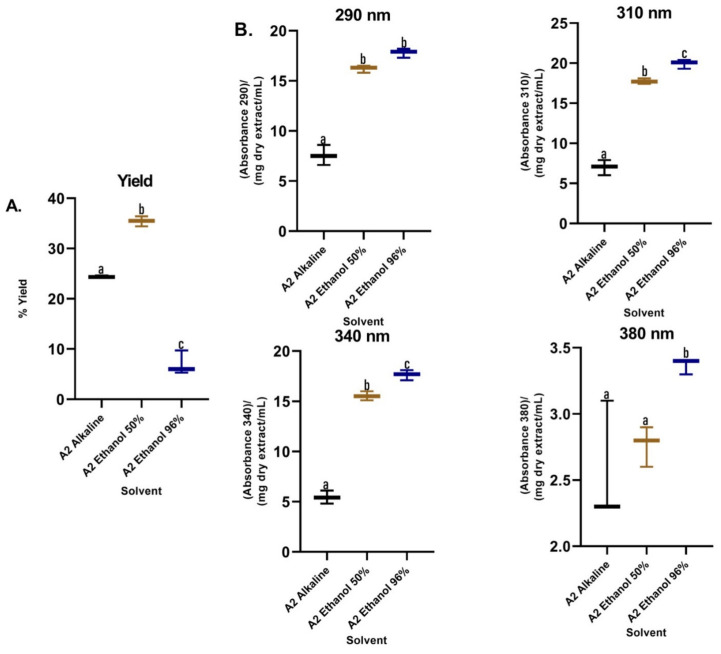
(**A**) Yields of *B. antioquensis* sequential extractions in different solvents. (**B**) Relative absorption coefficients (RAC) of extracts of *B. antioquensis* (Assay 2). Values with different letters are significantly different (*p* < 0.05).

**Figure 6 plants-12-00979-f006:**
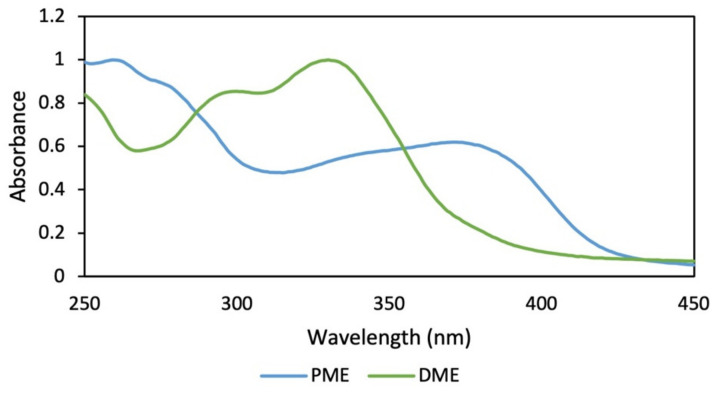
Absorption spectra of dry methanolic extract (DME) and hydrolyzed and purified methanolic extract (PME) of *B. antioquensis*.

**Figure 7 plants-12-00979-f007:**
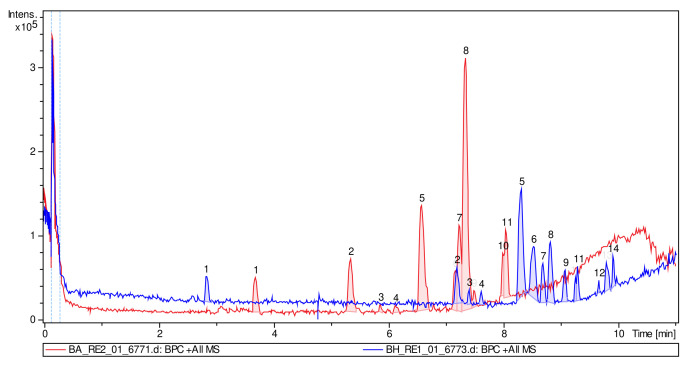
ESI–MS/MS total ion chromatogram profiles overlaid for DME (red) and PME (blue).

**Figure 8 plants-12-00979-f008:**
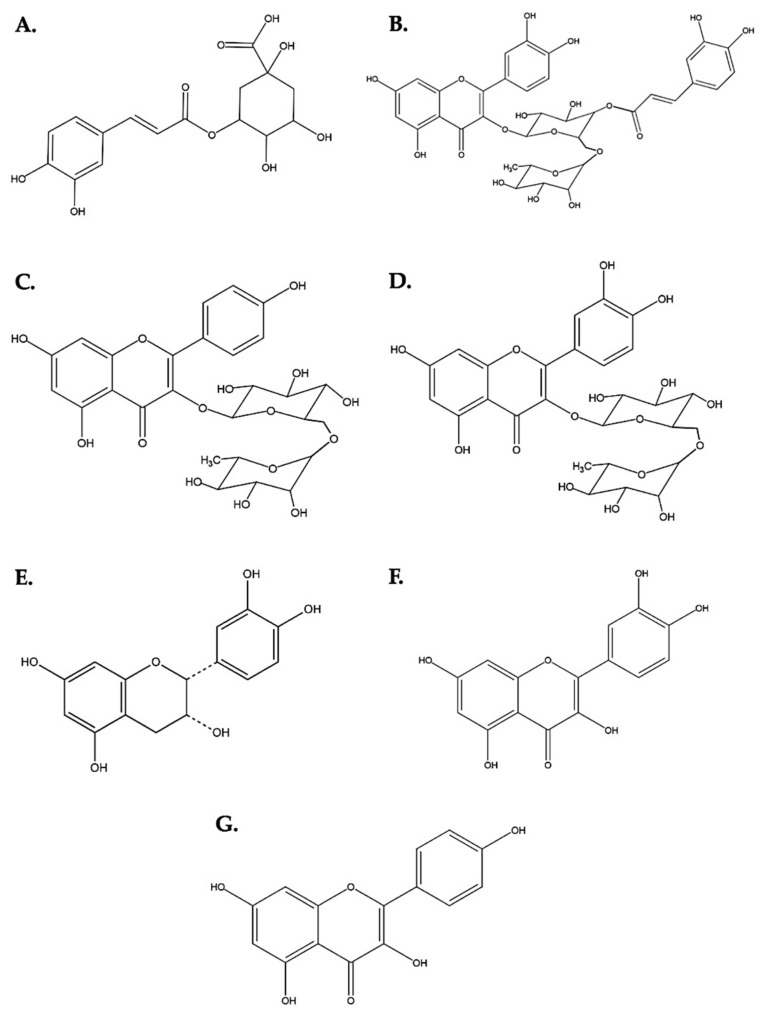
Chemical structures of the flavonoids present in DME and PME. (**A**) 3-O-caffeoylquinic acid (chlorogenic acid). (**B**) Quercetin-3-O-(4″′-O-caffeoyl)-rhamnopyranosyl-(1→6)-galactopyranose. (**C**) Kaempferol 3-O-rhamnopyranosyl-(1→6)-glucopyranose. (**D**) Quercetin 3-O-rhamnopyranosyl-(1→6)- glucopyranose (rutin). (**E**) Epicatechin. (**F**) Quercetin. (**G**) Kaempferol.

**Table 1 plants-12-00979-t001:** Total phenol content (TPC) and antiradical test.

Methodology	Solvent	TPC (mg GAE/g Extract)	EC_50_ (g Extract/mmol DPPH)
Assay 1	Absolute ethanol	206.6 ± 1.9 ^a^	0.31 ± 0.02 ^a^
	Methanol	248.6 ± 1.7 ^b^	0.24 ± 0.01 ^b^
Assay 2	Ethanol 96%	208.5 ± 5.1 ^a^	0.30 ± 0.03 ^a^
	Ethanol 50%	213.3 ± 4.7 ^c^	0.23 ± 0.01 ^b^
Assay 3	Methanol—DME	206.3 ± 4.6 ^a^	0.35 ± 0.02 ^a^
	Methanol—PME	417.6 ± 1.8 ^d^	0.11 ± 0.003 ^c^
Ascorbic acid	-	-	0.10 ± 0.001 ^c^

DME: Dry methanolic extract, PME: Hydrolyzed and purified methanolic extract. Results are expressed as the mean value ± standard deviation (*n* = 3). Values in the same column with different superscript letters are significantly different (*p* < 0.05).

**Table 2 plants-12-00979-t002:** Photoprotection and photostability parameters for formulations containing extracts and mixtures of extracts.

Photoprotection and Photostability Parameter	Exposure Time in the Solar Simulator				
	0 min	30 min	60 min	90 min	120 min
DME					
SPF*_in vitro_*	21.9 ± 3.3 ^a^	6.0 ± 0.6	5.0 ± 0.4	4.0 ± 0.4	4.0 ± 0.3 ^a^
%SPF*_effective_*	100	27.4	22.8	18.3	18.3
UVAPF	6.0 ± 0.5 ^a^	--	--	--	3.0 ± 0.4 ^a^
%UVAPF*_effective_*	100	--	--	--	50.0
$\lambda_{critical}$	374	378	379	380	381
UVA/UVB	0.702	0.757	0.784	0.798	0.803
SPF /UVAPF	3.6	--	--	--	1.3
PME					
SPF*_in vitro_*	9.4 ± 0.7 ^b^	8.0 ± 0.6	9.0 ± 0.6	9.0 ± 1.2	9.2 ± 1.0 ^b^
%SPF*_effective_*	100	85.1	95.7	95.7	97.9
UVAPF	8.0 ± 0.6 ^b^	--	--	--	8.0 ± 0.9 ^b^
%UVAPF*_effective_*	100	--	--	--	100
$\lambda_{critical}$	389	389	388	388	388
UVA/UVB	0.947	0.941	0.926	0.918	0.907
SPF /UVAPF	3.6	--	--	--	1.1
DME: PME (70:30)					
SPF*_in vitro_*	18.8 ± 1.4 ^a^	9.0 ± 0.6	8.0 ± 0.3	7.8 ± 0.4	7.8 ± 0.7 ^b^
%SPF*_effective_*	100	47.9	43.2	41.5	41.5
UVAPF	8.0 ± 0.4 ^b^	--	--	--	6.0 ± 0.3 ^c,e^
%UVAPF*_effective_*	100	--	--	--	75
$\lambda_{critical}$	383	384	384	385	385
UVA/UVB	0.812	0.841	0.852	0.856	0.861
SPF /UVAPF	2.3	--	--	--	1.3
DME: PME (80:20)					
SPF*_in vitro_*	21.0 ± 3.7 ^a^	9.0 ± 1.3	8.5 ± 1.1	8.0 ± 1.1	7.5 ± 1.2 ^b,c^
%SPF*_effective_*	100	42.8	40.5	38.1	35.7
UVAPF	5.0 ± 0.6 ^a^	--	--	--	5.0 ± 0.5 ^c^
%UVAPF*_effective_*	100	--	--	--	100
$\lambda_{critical}$	381	382	383	384	384
UVA/UVB	0.795	0.808	0.826	0.832	0.834
SPF /UVAPF	4.2	--	--	--	1.5

DME: Dry methanolic extract, PME: Hydrolyzed and purified methanolic extract. Results are expressed as the mean value ± standard deviation (*n* = 4). (--) The values of these parameters were not obtained at those times. Values in the same line followed by different letters are significantly different (*p* < 0.05).

**Table 3 plants-12-00979-t003:** Values of Biological Effective Protection Factors (BEPFs) for formulations containing extracts.

Sample	DME	PME	Positive Control (Sunscreen SPF 16)	Negative Control (Base Formulation)
Parameter				
**Biological effects mediated by UVA**				
**Elastosis**	5.6 ± 0.5 ^a^	10.9 ± 1.5 ^b^	11.0 ± 0.4 ^b^	1.0 ± 0.0 ^c^
**Photoaging**	8.5 ± 1.1 ^a^	11.3 ± 1.4 ^b^	28.7 ± 2.3 ^c^	1.0 ± 0.0 ^d^
**Lipid peroxidation**	6.1 ± 0.4 ^a^	7.6 ± 0.8 ^b^	13.9 ± 0.8 ^c^	1.0 ± 0.0 ^d^
**Biological effects mediated by UVB**				
**DNA damage**	10.5 ± 1.3 ^a^	6.1 ± 0.6 ^b^	13.9 ± 0.6 ^c^	1.0 ± 0.0 ^d^
**Photocarcinogenesis (NMSC)**	9.5 ± 1.1 ^a^	7.2 ± 0.7 ^b^	12.1 ± 0.6 ^c^	1.0 ± 0.0 ^d^
**Immunosuppression**	10.5 ± 1.4 ^a^	6.9 ± 0.7 ^b^	24.3 ± 2.6 ^c^	1.1 ± 0.0 ^d^
**Singlet oxygen production**	9.6 ± 1.3 ^a^	7.9 ± 0.8 ^a^	25.8 ± 2.8 ^b^	1.0 ± 0.0 ^c^

DME: Dry methanolic extract, PME: Hydrolyzed and purified methanolic extract. NMSC: non-melanoma skin cancer. Values in the same line followed by different letters are significantly different (*p* < 0.05).

## Data Availability

Not applicable.

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
