# Peer review of "Holistic Photoprotection, Broad Spectrum (UVA-UVB), and Biological Effective Protection Factors (BEPFs) from Baccharis antioquensis Hydrolysates Polyphenols"

_plants, 2023, doi:10.3390/plants12050979_

Round 1

Reviewer 1 Report

The authors of the manuscript titled 'Holistic photoprotection, broad spectrum (UVA-UVB) and Biological Effective Protection Factors (BEPFs) from Baccharis antioquensis hydrolysates polyphenols' presented a paper that draws attention to a current topic and the increasing interest in the use of natural extracts to replace physical and/or organic filters in sunscreen formulations. However, the work needs major revisions for it to take a form suitable for publication. There are several points to revise throughout the manuscript, and a few pointers follow that I hope will help the authors to make the right changes.

All considerng that I COULD NOT VIEW THE SUPPLEMENTARY MATERIAL as it was not attached as a file to the review.

Line 46-49: the sentence is a bit confusing...I suggest the authors rewrite it in a simpler and more straightforward manner.

Line 71: TiO2-nano must be replaced with TiO2-nano

Line 68-75: Sentence to be revised and edited, it is too long and the sense is lost.

Linea 75-79: revise the sentence, it is incomplete or there is something grammatical about it that makes it seem so.

Line 173: utin must be replaced with Rutin

Paragraph 2.2.1. HPLC-DAD analysis: The authors' statements on lines 173-176 need to be revised. In HPLC one cannot identify a substance on the basis of similarity of profile and similar retention time, this applies to both Rutin and Quercetin. Either the authors make an exact assignment of the peaks (e.g. by addition of the standard to the sample and relative peak accretion), otherwise I would postpone the identification to the UPLC-MS analysis in the next paragraph.

Line 219: On what basis was the range of extract concentration to be tested chosen? considering that it is a range far away from the 10% concentration used in the formulation.

Line 226-227: the authors state “cosmetic emulsions containing 10 of extract or mixture of extracts”, but 10 what? Complete with the missing unit of measurement (grams I presume).

Line 230-232: how was the stability of the emulsions determined? were precise parameters assessed or any particular tests carried out?

Line 254-255: The authors affirm “Other photoprotection parameters did not showed significant changes with prolonged  irradiation”, but actually, for the critical lambda of the DME-containing formulation, a variation of as much as 7 nm is observed between T0 and after 2 hours. This means that something has changed in the spectrum, which is also supported by the instability of the filter parameters.

Line 257: change the caption of Table 2, which refers to formulations containing extracts, not to extracts (or at least in the text the authors state this).

Table 2: Why were not all the parameters in the table measured at the various irradiation times?

Table 3: it is unclear whether the authors refer to the extracts in formulation or to the extracts as such

Line 304-306: the extract that is richest in polyphenols and has activity values comparable to ascorbic acid is PME, not DME; from the sentence, DME seems to be the better performing extract.

Line 311-314: I suggest the authors replace “the results are consistent with what reported by Mejía-Giraldo et al., 2016 [22]. This evidence added to that reported by Mejía-Giraldo et al., 2016 [22], suggest the presence of flavonoids, which is confirmed by the mass spectrometry results.” with “results are consistent with what reported by Mejía-Giraldo et al., 2016 [22], suggesting the presence of flavonoids, confirmed by mass spectrometry results.

DISCUSSION:

Line 323-326: The authors compare the cytotoxicity data obtained on HaCaT cells with the IC50 value obtained in the 2016 work on U937 cells, which are a pro-monocytic, human myeloid leukaemia cell line, concluding that hydrolysis does not increase the cytotoxicity of the extract. In fact, with an IC50 of 44.2 μg/mL, the extract was more cytotoxic for the healthy line than the tumour line.

I suggest that the authors list in the discussion (lines 327-332) the cell lines used in the different works to calculate IC50s.

Line 444: the authors declare “expressed as milligrams of gallic acid equivalents (GAE)”, mentre in Tabella 1 l’unità di misura è mg AGE/g extract

Section 4.5.3: generally an IC50 value is expressed as μg/mL, also for the DPPH test. I find the unit of measurement reported by the authors less straightforward, but that is just my opinion.

Section 4.5.4.: I suggest the authors review the gradient of analysis written at lines 464-467. This paragraph suggests a chromatographic run of 7 minutes, when the text (paragraph 2.2.1.) speaks of retention times exceeding 11 minutes.

Section 4.6: The authors assert that the cells were incubated with extract concentrations of 3.9 to 60 μg/mL, whereas at line 219 they had claimed a range of 10-60 μg/mL.

Author Response

Response to Reviewer 1

  • The authors of the manuscript titled 'Holistic photoprotection, broad spectrum (UVA-UVB) and Biological Effective Protection Factors (BEPFs) from Baccharis antioquensishydrolysates polyphenols' presented a paper that draws attention to a current topic and the increasing interest in the use of natural extracts to replace physical and/or organic filters in sunscreen formulations. However, the work needs major revisions for it to take a form suitable for publication. There are several points to revise throughout the manuscript, and a few pointers follow that I hope will help the authors to make the right changes.
  • All considering that I COULD NOT VIEW THE SUPPLEMENTARY MATERIAL as it was not attached as a file to the review.

Supplementary material was sent from the first submission.

  • Line 71: TiO2-nano must be replaced with TiO2-nano

The change was made.

  • Line 68-75: Sentence to be revised and edited, it is too long and the sense is lost.

The text was completely rewritten.

  • Linea 75-79: revise the sentence, it is incomplete or there is something grammatical about it that makes it seem so.

The text was completely rewritten.

  • Line 173: utin must be replaced with Rutin

The change was made.

  • Paragraph 2.2.1. HPLC-DAD analysis: The authors' statements on lines 173-176 need to be revised. In HPLC one cannot identify a substance on the basis of similarity of profile and similar retention time, this applies to both Rutin and Quercetin. Either the authors make an exact assignment of the peaks (e.g. by addition of the standard to the sample and relative peak accretion), otherwise I would postpone the identification to the UPLC-MS analysis in the next paragraph.

The reviewer is right. The text was completely rewritten.

  • Line 219: On what basis was the range of extract concentration to be tested chosen? considering that it is a range far away from the 10% concentration used in the formulation.

For this test, the minimum amount of extract that will present cytotoxic activity was considered (IC50). In addition, it should be considered that cytotoxicity tests are the first approach to the evaluation of the safety of a pharmaceutical or cosmetic ingredient, which is done with cell cultures in suspension, cells that do not have all the defense mechanisms or barrier function intact that human skin has. After this it is necessary to do dermatological tests

  • Line 226-227: the authors state “cosmetic emulsions containing 10 of extract or mixture of extracts”, but 10 what? Complete with the missing unit of measurement (grams I presume).

The change was made.

  • Line 230-232: how was the stability of the emulsions determined? were precise parameters assessed or any particular tests carried out?

The preliminary stability of the emulsion was determined by subjecting the formulations to the following conditions:

Centrifuge test: 3000 rpm for 30 minutes.

Temperature cycles: formulations were maintained for 24 hours at high temperature (40°C +/- 5°C), then 24 hours at room temperature (25°C+/- 5°C) and 24 hours under refrigeration (8ºC+/ -2°C).

The parameters evaluated were:

Organoleptic: without significant changes in appearance, texture, color, or odor.

Physicochemicals: phase separation and pH.

Previous assays leading to the formulation were not included in the publication.

  • Line 254-255: The authors affirm “Other photoprotection parameters did not showed significant changes with prolonged  irradiation”, but actually, for the critical lambda of the DME-containing formulation, a variation of as much as 7 nm is observed between T0 and after 2 hours. This means that something has changed in the spectrum, which is also supported by the instability of the filter parameters.

This is correct, there is a 7 nm shift in the critical wavelength. However, considering that the critical wavelength (λc) for the test product is defined as the wavelength at which the area under the absorbance spectrum for the irradiated product starting at 290 nm at λc is 90 % of the integral of the absorbance spectrum from 290 nm to 400 nm, and that in this and in the majority of photoprotection products the absorption spectrum is the result of the contribution of different compounds in the sample, it is difficult to determine the instability of the filter mix, as this is not a maximum absorbance peak. In addition, as observed in figure 6, the DME spectrum shows a significant decay above 375 nm, therefore, this change in λc can also be due to a change in absorbance.

  • Line 257: change the caption of Table 2, which refers to formulations containing extracts, not to extracts (or at least in the text the authors state this).

The change was made.

  • Table 2: Why were not all the parameters in the table measured at the various irradiation times?

According to ISO 24443, numeral 5.7.3 and 5.10, UVAPF calculation is based on the initial value (UVAPF0) and the final value, after irradiation, for which, from experimental point of view, it is not possible to calculate the UVAPF, or parameters derived from it at each intermediate time.

  • Table 3: it is unclear whether the authors refer to the extracts in formulation or to the extracts as such.

Thank you very much for the recommendation. The change was made.

  • Line 304-306: the extract that is richest in polyphenols and has activity values comparable to ascorbic acid is PME, not DME; from the sentence, DME seems to be the better performing extract.

Thank you very much for the recommendation. The change was made.

  • Line 311-314: I suggest the authors replace “the results are consistent with what reported by Mejía-Giraldo et al., 2016 [22]. This evidence added to that reported by Mejía-Giraldo et al., 2016 [22], suggest the presence of flavonoids, which is confirmed by the mass spectrometry results.” with “results are consistent with what reported by Mejía-Giraldo et al., 2016 [22], suggesting the presence of flavonoids, confirmed by mass spectrometry results.

 Thank you very much for the recommendation. The change was made.

DISCUSSION:

  • Line 323-326: The authors compare the cytotoxicity data obtained on HaCaT cells with the IC50 value obtained in the 2016 work on U937 cells, which are a pro-monocytic, human myeloid leukaemia cell line, concluding that hydrolysis does not increase the cytotoxicity of the extract. In fact, with an IC50 of 44.2 μg/mL, the extract was more cytotoxic for the healthy line than the tumour line. I suggest that the authors list in the discussion (lines 327-332) the cell lines used in the different works to calculate IC50s.

The information about the cell lines used in the references have been included in the manuscript.

  • Line 444: the authors declare “expressed as milligrams of gallic acid equivalents (GAE)”, mentre in Tabella 1 l’unità di misura è mg AGE/g extract.

 The change was made in the Table 1.

  • Section 4.5.3: generally, an IC50 value is expressed as μg/mL, also for the DPPH test. I find the unit of measurement reported by the authors less straightforward, but that is just my opinion.

Thank you very much for the recommendation.

  • Section 4.5.4.: I suggest the authors review the gradient of analysis written at lines 464-467. This paragraph suggests a chromatographic run of 7 minutes, when the text (paragraph 2.2.1.) speaks of retention times exceeding 11 minutes.

The change was made, gradient times were adjusted.

  • Section 4.6: The authors assert that the cells were incubated with extract concentrations of 3.9 to 60 μg/mL, whereas at line 219 they had claimed a range of 10-60 μg/mL.

The values on the manuscript have been corrected. The values tested varied from 3.9 to 60 μg/mL.

Reviewer 2 Report

In this paper the authors propose a dermocosmetic formulation obtained from the extract of Baccharis antioquensis.  They isolated several molecules from the methanolic extract and use a battery of test to check distinct pharmacological properties, including antiradical capacity, UVA-UVB photoprotection, photoaging, immunosuppression, and DNA damage.  The paper is quite interesting, but it needs some revision before it is accepted for publication.

Remove Figure 1 from the introduction.  It does not add anything to the section, and it will need a lot of explanation to support the data.

Please check this, it is not correct. “Values in the same column followed by different letters are significantly different at the 5 % level”.  Level of what?  p-value?  Use what you wrote on Statistical analysis.

Improve the resolution of molecules in Figure 8.

In Figure 9, please use bars, showing the error bars as well.  That type of graph is not common.  Did you test all those concentrations?

In Table 2, explain the meaning of “--“.  The same for the letters within the Table, a, b, c…

Statistical analysis.  Please remember that ANOVA works on parametric data.  Did you check that?

I really do not like the approach to measure BEPFs.  Be sure to include at least three references where it is used. Please explain how to interpret the obtained results.  For me, to measure DNA damage just from the spectra is insane. The authors should be very careful when discussing on this issue.  For instance, in Table 3, DNA damage from the Positive Control is 13.9, and from the extracts 10.5 and 6.1.  Thus, which one induces more DNA damage?  Should I suppose that the positive control induces DNA damage when it is a chemical used to control DNA damage induced by UV light?  This issue has no single sentence in the Discussion Section, although it is a very important one. 

The authors should also mention how they develop the formulations.  Were they from literature or just a guess?

In Table 2, minutes refer to exposure? Reaction?

Abstract: …among others.

The word “etc” should be avoided in scientific papers.

Are you sure the yields for Absolute ethanol and A1 methanol are greater than 10%?  Please check. 

Author Response

Response to Reviewer 2

  • In this paper the authors propose a dermocosmetic formulation obtained from the extract of Baccharis antioquensis.  They isolated several molecules from the methanolic extract and use a battery of test to check distinct pharmacological properties, including antiradical capacity, UVA-UVB photoprotection, photoaging, immunosuppression, and DNA damage.  The paper is quite interesting, but it needs some revision before it is accepted for publication.

Thank you for your comments and recommendations. The review was carried out.

  • Remove Figure 1 from the introduction.  It does not add anything to the section, and it will need a lot of explanation to support the data.

Thanks for the recommendation. The authors consider that the information in the graph is relevant to understand the different biological effects, therefore an explanatory text is added to the manuscript.

  • Please check this, it is not correct. “Values in the same column followed by different letters are significantly different at the 5 % level”.  Level of what?  p-value?  Use what you wrote on Statistical analysis.

The change was made.

  • Improve the resolution of molecules in Figure 8.

The change was made in the Figure 8.

  • In Table 2, explain the meaning of “--“.  The same for the letters within the Table, a, b, c…

The explanation was added.

  • Statistical analysis.  Please remember that ANOVA works on parametric data.  Did you check that?

The data were tested and had a normal distribution.

  • I really do not like the approach to measure BEPFs.  Be sure to include at least three references where it is used. Please explain how to interpret the obtained results.  For me, to measure DNA damage just from the spectra is insane. The authors should be very careful when discussing on this issue.  For instance, in Table 3, DNA damage from the Positive Control is 13.9, and from the extracts 10.5 and 6.1.  Thus, which one induces more DNA damage?  Should I suppose that the positive control induces DNA damage when it is a chemical used to control DNA damage induced by UV light?  This issue has no single sentence in the Discussion Section, although it is a very important one. 

The BEPFs were reported by De la Coba et al (2019) to offer new protection factors in addition to SPF (solar protection factor) and UVPF and it is based in the same approach and photobiological basis on the calculation of SPF as it is explained in de la Coba et al. (2019) paper

“Currently, the erythema (UVB-mediated skin injury) action spectrum is used for the in vitro SPF

determinations, and Persistent Pigment Darkening (PPD) action spectrum for the UVA protection

factor (UVA-PF). The European Union recommends both a critical wavelength of more than 370 nm

and a UVA–PF of at least one third of the labelled SPF as the criterion for labelling as either UVA

or broad-spectrum protection (ISO 24443:2012) [11]. However, other acute damages and most of the

chronic ones are mediated by other wavelengths in the UV range, with known action spectra which

could be used for complementary evaluations of sunscreen testing and labelling of broad-spectrum

photoprotective capability in vitro. Especially, it may help to differentiate the effectiveness of sunscreen

formulations with similar SPF in the future. Some of these action spectra are DNA damage [12],

photocarcinogenesis non melanoma skin cancer (NMSC) [13], systemic immunosuppression of

contact hypersensitivity (CHS) [14], cis-photoisomerization of urocanic acid [15], formation of oxygen

radicals such as singlet oxygen [16] and photoaging [17], all related to UVB and UVA radiation

exposure-induced skin damage”

  1. International Organization for Standardization. ISO 2443: 2012—Determination of Sunscreen UVA Photoprotection In Vitro; ISO: Geneva, Switzerland, 2012.
  2. Setlow, R.B. The wavelengths in sunlight effective in producing skin cancer: A theoretical analysis. Proc.

Natl. Acad. Sci. USA 1974, 71, 3363–3366. [CrossRef] [PubMed]

  1. De Gruijl, F.R.; Van der Leun, J.C. Estimate of the wavelength dependency of ultraviolet carcinogenesis

in humans and its relevance to the risk assessment of a stratospheric ozone depletion. Health Phys. 1994,

67, 319–325. [CrossRef] [PubMed]

  1. De Fabo, E.; Noonan, F. Mechanism of immune suppression by ultraviolet irradiation in vivo. I. Evidence

for the existence of a unique photoreceptor in skin and its role in photoimmunology. J. Exp. Med. 1983,

158, 84–98. [CrossRef] [PubMed]

  1. McLoone, P.; Simics, E.; Barton, A.; Norval, M.; Gibbs, N.K. An action spectrum for the production of

cis-urocanic acid in human skin in vivo. J. Investig. Dermatol. 2005, 124, 1071–1074. [CrossRef] [PubMed]

  1. Hanson, K.M.; Simon, J.D. Epidermal trans-urocanic acid and the UV-A-induced photoaging of the skin.

Proc. Natl. Acad. Sci. USA 1998, 95, 10576–10578. [CrossRef] [PubMed]

  1. Bissett, D.L.; Hannon, D.P.; Orr, T.V. Wavelength dependence of histological, physical, and visible changes in

chronically UV-irradiated hairless mouse skin. Photochem. Photobiol. 1989, 50, 763–769. [CrossRef] [PubMed]

As it is explained in the methodology of de la Coba et al (2019) paper, the calculation of BEPFs is done as the SPF calculation. It is based in the action spectra of biological effects. In the case of SPF is based on the action spectra of erythema produced in the UVB region on the spectra as other biological effects included in BEPFs as DNA damage, photocarcinogenesis (Non melanoma skin cancer) and photosiomerization of urocanic acid whereas there are other biological effect produced in the UVA region of the spectra as formation of singlet oxygen and photoaging as it is explained in material and methods of the De la Coba et al (2019) manuscript.

Methodology f calculation of SPF and BEPFs (de la Coba et al., 2019)

In vitro Sun Protection Factor (SPF) Determination

SPF was characterized according to the Diffey method [96] with some modifications, and protection in the UVA band was calculated by the critical wavelength [97,98]. The UVA/UVB ratio defines the performance of a sunscreen in the UVA range (320–400 nm) in relation to its performance in the UVB range (290–320 nm), while the critical wavelength is given as the upper limit of the spectral range from 290 nm on, within which 90% of the area under the extinction curve of the whole UV range between 290 and 400 nm is covered. When the critical wavelength is 370 nm or greater, the product is considered broad spectrum, which denotes balanced protection throughout the UVB and UVA ranges.

Transmittance measurements were performed on 3M surgical Transpore self-standing substrate, a material chosen to simulate the roughness of human skin with better correlation between in vitro and

in vivo SPF measurements when the test is not pre-irradiated [99]. Transpore tape was stuck onto a light-diffusing double- ground quartz plate and placed on an analytical balance. The sunscreens were plotted onto the tape from a pre-weighted syringe at a rate of 2 mg_cm?2 along the selected surface (4 cm2) and spread evenly with a gloved finger for about 30 s. The preparation was allowed to dry for 15 min and then, was irradiated using a double band UV spectrophotometer (Shimadzu UV-1800) between 290–400 nm wavelengths, collecting the resulting transmittance spectrum. A Transpore

substrate with base formula was used as a baseline.

  1. Diffey, B.L.; Robson, J. A new substrate to measure sunscreen protection factors throughout the ultraviolet

spectrum. J. Soc. Cosmet. Chem. 1989, 40, 127–133.

  1. Diffey, B.L. Indices of protection from in vitro assay of sunscreens. Sunscreens Dev. Eval. Regul. Asp. 1997, 589–600.
  2. Diffey, Bl. A method for broad spectrum classification of sunscreens. Int. J. Cosmet. Sci. 1994, 16, 47–52.

[CrossRef]

  1. Garoli, D.; Pelizzo, M.G.; Nicolosi, P.; Peserico, A.; Tonin, E.; Alaibac, M. Effectiveness of different substrate

materials for in vitro sunscreen tests. J. Dermatol. Sci. 2009, 56, 89–98. [CrossRef]

The sun protection factor (SPF) was calculated from transmission measurements according to:

where

El = CIE standard Skin reference erythema action spectrum [77].

Tl = Transmitance values (0–1).

Sl = Spectral irradiance of a midday clear sky of summer, terrestrial sunlight in Spain

Sunlight was measured by using a portable double monochromator spectrorradiometer Bentham

IDR300-PSL (Bentham Co., Reading, UK) which takes measurements inWm-2 s-1 between 200 to

500 nm (Device located at Unit of Photobiology of the Central Service for Research Support, SCAI,

University of Malaga, Malaga, Spain).

Biological Effective Protection Factors (BEPFs)

BEPF calculation for a determined UV-mediated skin response was obtained through transmission measurements according to the Diffey method using the relative action spectrum for this biological effect (Table 4). BEPF can be understood as an indicator given to sunscreen consumers, informing them about the number of times a photoprotected person can be exposed versus a person with unprotected

skin before this process begins to develop.

In this study, the level of UV photoprotection for the different formulated emulsions was determined for a total of seven UV-biological mediated processes with a well-known action spectra associated with human skin. Some of them have maximum effect on UVB (UVB-BEPFs) and others in the UVA wavelengths (UVA-BEPFs) (Table 4). Data for erythema, DNA damage, and photocarcinogenesis of non-melanoma skin cancers (NMSC- SCUP-h) spectra were available in the literature [12–14]. For the other action spectra selected, cubic splinic interpolation between the data points of the respective action spectrum has been employed to provide values of 1 nm increments and the integral in the Equation (1) replaced by a summation in 1 nm step. Splinic interpolation was done by means of the software Table curve 2D 5.0.1. (Systat Software Inc., San Jose, CA, USA). The error in the interpolation and summation in 1 nm step is estimated to be lower than 5% (Figure 5).

  •  
  1. Setlow, R.B. The wavelengths in sunlight effective in producing skin cancer: A theoretical analysis. Proc.

Natl. Acad. Sci. USA 1974, 71, 3363–3366. [CrossRef] [PubMed]

  1. De Gruijl, F.R.; Van der Leun, J.C. Estimate of the wavelength dependency of ultraviolet carcinogenesis

in humans and its relevance to the risk assessment of a stratospheric ozone depletion. Health Phys. 1994,

67, 319–325. [CrossRef] [PubMed]

  1. De Fabo, E.; Noonan, F. Mechanism of immune suppression by ultraviolet irradiation in vivo. I. Evidence

for the existence of a unique photoreceptor in skin and its role in photoimmunology. J. Exp. Med. 1983,

158, 84–98. [CrossRef] [PubMed]

  • For instance, in Table 3, DNA damage from the Positive Control is 13.9, and from the extracts 10.5 and 6.1.  Thus, which one induces more DNA damage?

The values of the different BEPFs (Table 3) must be interpreted as SPF. As we ex-plained above BEPFs are indicator of photoprotection of different biological effect. Thus, the value of 13.9 in the positive control is a protection factor higher than that from the ex-tracts 10.5 and 6.1 as SPF 50 means higher protection against erythema than SPFs values of 15 or 30. All values of BEPFs of positive control (sunscreen SPF 16) are higher than pro-tection of the extracts DME or PME, except in the case of elastosis that is the same in the case of PME extracts and positive control. DME extract presented higher photoprotection than that of PME extracts against biological effects mediated by UVB radiation (Immuno-supression, DNA damage, photocarcinogenesis (NMSC)), due to DME extracts presented the maximal absorption in the UVB region of the spectra. Whereas PME extracts presented higher photoprotection than that of DME extracts against biological effects mediated by UVA radiation as elastosis, photoaging, and lipid peroxidation, due to PME extract pre-sented maximal absorption in the UVA region spectra. Thus, a cream containing both DME and PME could be considered a broad band UV screen product, since it can protect against biological effects mediated by both UVB and UVA.

This paragraph is added to the discussion.

  • The authors should also mention how they develop the formulations.  Were they from literature or just a guess?

The development of the formulation was part of the collaborative work between the two research groups, based on the previous knowledge in natural cosmetic formulations that the research group from the University of Malaga had. Within the work, the behavior of the formulation was evaluated with the addition of the different extracts and their mixtures.

The preliminary stability of the emulsion was determined by subjecting the formulations to the following conditions:

Centrifuge test: 3000 rpm for 30 minutes.

Temperature cycles: formulations were maintained for 24 hours at high temperature (40°C +/- 5°C), then 24 hours at room temperature (25°C+/- 5°C) and 24 hours under refrigeration (8ºC+/ -2°C).

The parameters evaluated were:

Organoleptic: without significant changes in appearance, texture, color, or odor.

Physicochemicals: phase separation and pH.

Previous assays leading to the formulation were not included in the publication.

  • In Table 2, minutes refer to exposure? Reaction?

This time is referred to exposure time in the solar simulator. This was specified in Table 2.

Abstract: …among others.

The change was made.

  • The word “etc” should be avoided in scientific papers.

It was removed from the entire manuscript.

  • Are you sure the yields for Absolute ethanol and A1 methanol are greater than 10%?  Please check. 

Yes, the authors are sure of the performance of these solvents since it was a test where successive extractions were carried out to obtain enough dry extract to carry out subsequent tests. This plant gives very good yields and on several occasions, we have published these results, even with higher yields.

Reviewer 3 Report

The develop a dermocosmetic formulation with broad-spectrum photoprotection from the hydrolysates and purified polyphenols obtained from the endemic Colombian high-mountain plant Baccharis antioquensis, is reported in this manuscript. Hence, the extraction of its polyphenols with different solvents was evaluated, followed by hydrolysis and purification, in addition to the characterization of its main compounds by HPLC-DAD and HPLC-MS. Its photoprotective capacity through the measurement of the Sun Protection Factor (SPF), UVA Protection Factor (UVAPF), other Biological Effective Protection Factors (BEPFs) and its safety through the cytotoxicity, was also evaluated.

This work is interesting, and it will attract the attention scientist who work in the field. The manuscript is well written, and the conclusions supported by the experiments. The manuscript can be published in MDPI Plants after minor revision.

This work shows high similarity with the previous one reported by the same group ( Antioxidants 2021, 10, 1904 and Photochemistry and Photobiology, 2016, 92: 742–752; ibid, 2016, 92: 150–157; ibid 2022, 98: 211–219 ). The novelty in this work, in respect with those previous reported should be mentioned.

Moreover extended common text parts in the experimental section should eliminated e.g. 4.5.2.Total phenol content from  Photochemical & Photobiological Sciences (2021) 20:1585–1597, 4.5.3. Evaluation of the inhibition capacity of DPPH, 4.5.3. Evaluation of the inhibition capacity of DPPH from Photochemistry and Photobiology, 2022, 98: 211–219 etc.

Cytotoxicity tests. the number of HaCaT cells per well should be reported

Author Response

Response to Reviewer 3

  • The develop a dermocosmetic formulation with broad-spectrum photoprotection from the hydrolysates and purified polyphenols obtained from the endemic Colombian high-mountain plant Baccharis antioquensis, is reported in this manuscript. Hence, the extraction of its polyphenols with different solvents was evaluated, followed by hydrolysis and purification, in addition to the characterization of its main compounds by HPLC-DAD and HPLC-MS. Its photoprotective capacity through the measurement of the Sun Protection Factor (SPF), UVA Protection Factor (UVAPF), other Biological Effective Protection Factors (BEPFs) and its safety through the cytotoxicity, was also evaluated.

This work is interesting, and it will attract the attention scientist who work in the field. The manuscript is well written, and the conclusions supported by the experiments. The manuscript can be published in MDPI Plants after minor revision.

Thank you for your comments and recommendations. The review was carried out.

  • This work shows high similarity with the previous one reported by the same group (Antioxidants 2021, 10, 1904 and Photochemistry and Photobiology, 2016, 92: 742–752; ibid, 2016, 92: 150–157; ibid 2022, 98: 211–219). The novelty in this work, in respect with those previous reported should be mentioned.

Thanks for the recommendation it was considered. The novelty of this work was the hydrolysis of the main components of the extract, and its purification, to improve photostability. In addition, the inclusion in a new formulation of natural cosmetics and the comprehensive evaluation of its photoprotective efficacy. This is mentioned in the abstract and in the discussion.

  • Moreover, extended common text parts in the experimental section should eliminated e.g., 4.5.2. Total phenol content from Photochemical & Photobiological Sciences (2021) 20:1585–1597, 4.5.3. Evaluation of the inhibition capacity of DPPH, 4.5.3. Evaluation of the inhibition capacity of DPPH from Photochemistry and Photobiology, 2022, 98: 211–219 etc.

Thanks for the recommendation it was considered.

  • Cytotoxicity tests. the number of HaCaT cells per well should be reported

It was used 6000 cells/well. This information has been included in the manuscript as requested.

Reviewer 4 Report

The content of this paper was interesting and easy to read. The paper evaluated the compounds extracted from Baccharis antioquensis based on solid experimental data, and can provide useful information for interested researchers. The content seems to match the scope of the journal submitted. I'm not an English-speaking person, but there were some sentences that I thought were strange. For example, I don't think the noun "hydrolysate" on line 146 applies here. Please recheck your manuscript and try to eliminate such mistakes.

Author Response

Response to Reviewer 4

  • The content of this paper was interesting and easy to read. The paper evaluated the compounds extracted from Baccharis antioquensisbased on solid experimental data and can provide useful information for interested researchers. The content seems to match the scope of the journal submitted. I'm not an English-speaking person, but there were some sentences that I thought were strange. For example, I don't think the noun "hydrolysate" on line 146 applies here. Please recheck your manuscript and try to eliminate such mistakes.

Thank you for your comments and recommendations. The review was carried out.

Round 2

Reviewer 1 Report

The authors have responded thoroughly to all the reviewer's requests and suggestions. Therefore, I now believe that the work is worthy of publication.

Reviewer 2 Report

I insist that Figure 1 should be removed from the introduction.  Send it to Supplementary material and add a credible explanation.  The one present REALLY does not support much.  Trying to obtain DNA damage data from UV spectra is insane.  This could be an exercise, but should not be taken as something “real”.  Remember, DNA damage, and the other physiological effects, will depend on metabolic activation, and many other features of the organism.  Moreover, it is an extract, a mixture. The extrapolation would be a lot more difficult.  Detail this on the Discussion.

When answering to the reviewer, please show the changes. Writing something such as the “explanation was added” is not enough.

“The data were tested and had a normal distribution”.  This has to be written in the document. 

Check the English again in the paragraph added to the Discussion Section. Still, it clarifies a bit the point on comparing BEPFs but real Discussion is missing.

This paper should have a certified English revision.

They should remake figure 7. Numbers in x-axis are not clear.  The same the legend in y-axis. Please edit it.  You can make in professional. 

Reviewer 4 Report

The authors responded to all reviewers' comments.